# Endovascular Treatment of Basilar Artery Occlusion: What Can We Learn from the Results?

**DOI:** 10.3390/medicina59010096

**Published:** 2022-12-31

**Authors:** Aleksandra Ekkert, Une Jokimaitytė, Valerija Tutukova, Givi Lengvenis, Rytis Masiliūnas, Dalius Jatužis

**Affiliations:** 1Centre of Neurology, Vilnius University, LT-08661 Vilnius, Lithuania; 2Klaipėda Seamen’s Hospital, LT-92288 Klaipėda, Lithuania

**Keywords:** posterior circulation stroke, vertebrobasilar stroke, endovascular treatment, bridging therapy, basilar artery occlusion

## Abstract

*Background and Objectives*: Current guidelines lack specific endovascular treatment (EVT) recommendations for posterior circulation stroke (PCS). The results of earlier studies are controversial. We aimed to compare early hospital outcomes of stroke caused by large-vessel occlusion (LVO) treated with EVT or bridging therapy (BT) in anterior circulation stroke (ACS) versus PCS (middle cerebral artery occlusion (MCAO) and basilar artery occlusion (BAO), and establish the risk factors for poor outcome. *Materials and Methods*: we analyzed the data of 279 subjects treated with EVT due to LVO-caused stroke in a comprehensive stroke centre in 2015–2021. The primary outcome was hospital mortality, secondary outcomes were National Institutes of Health Stroke Scale (NIHSS) after 24 h, early neurological deterioration, futile recanalization (FR), the ambulatory outcome at discharge, and complications. *Results*: BAO presented with higher baseline NIHSS scores (19 vs. 14, *p* < 0.001), and longer door-to-puncture time (93 vs. 82 min, *p* = 0.034), compared to MCAO. Hospital mortality and the percentage of FR were the same in BAO and almost two times higher than in MCAO (20.0% vs. 10.3%, *p* = 0.048), other outcomes did not differ. In BAO, unsuccessful recanalization was the only significant predictor of the lethal outcome, though there were trends for PAD and RF predicting lethal outcome. A trend for higher risk of symptomatic intracranial hemorrhage (sICH) was observed in the BAO group when BT was applied. Nevertheless, neither BT nor sICH predicted lethal outcomes in the BAO group. *Conclusions*: Compared to the modern gold standard of EVT in the ACS, early outcomes in BAO remain poor, there is a substantial amount of FR. Nevertheless, unsuccessful recanalization remains the strongest predictor of lethal outcomes. BT in PCS might pose a higher risk for sICH, but not the lethal outcome, although this finding requires further investigation in larger trials.

## 1. Introduction

Vertebrobasilar or posterior circulation stroke (PCS) accounts for 5–19% of all ischemic strokes treated with reperfusion therapy (RT) [1]. Posterior circulation anatomy, hemodynamics, and thrombus composition differ from the aforementioned parameters in the anterior circulation [2,3]. Moreover, PCS manifests more frequently with atypical and non-focal neurological symptoms, resulting in belated diagnosis and treatment [4,5]. Given this, the statement that PCS might require a customized workup and treatment strategy seems plausible.

While PCS is presented by a broad spectrum of clinical phenotypes, particular attention is drawn to the subgroup of cerebral infarctions caused by basilar artery occlusion (BAO). Despite accounting for only 1% of all ischemic stroke cases, it frequently results in devastating stroke course, poor functional outcomes, and high mortality, especially when reperfusion therapy is not applied [6,7,8]. Subjects presenting with PCS were excluded from the majority of trials dedicated to ischemic stroke endovascular treatment (EVT) [9,10,11,12]. The latest EVT recommendations for PCS provided by European Stroke Organisation (ESO)—European Society for Minimally Invasive Neurological Therapy (ESMINT) Guidelines on Mechanical Thrombectomy in Acute Ischemic Stroke (further referred to as “ESO-ESMINT guidelines”) are based on ACS treatment results [13], making unclear if current RT strategies can be generalized to PCS patients.

Some doubt still exists regarding the effectiveness of EVT in BAO due to insufficient evidence. Although Acute Basilar Artery Occlusion: Endovascular Interventions vs. Standard Medical Treatment (BEST) and Basilar Artery International Cooperation Study (BASICS) trials did not show a clear benefit of EVT in BAO, recent results from Endovascular Treatment for Acute Basilar Artery Occlusion (ATTENTION) trial proved that EVT can be effective in BAO within 12 h [14,15].

Some authors state that the outcomes of EVT in ACS and PCS do not differ significantly [16]. In the case of BAO, some report higher rates of futile recanalization and subsequent 90-day mortality compared to middle cerebral artery occlusion (MCAO), while others highlight comparable outcomes [17,18].

Recanalization is the most credible predictor of good outcomes, including survival rates and functional independence [7,19,20,21]**.** Other predictors of a good outcome are shorter time from stroke onset [19,21], younger age [22], and lower baseline National Institutes of Health Stroke Scale (NIHSS) scores [23]. Extensive baseline ischemia, ventilation support, history of atrial fibrillation, previous stroke [20], hypertension and diabetes mellitus [24], and EVT using stent-retriever or combined aspiration and stent-retriever approach (compared to aspiration only) [23] are associated with the risk of poor outcome.

It is worth noting that the majority of previous studies on this topic were published before or in 2019. That means that new ESO-ESMINT recommendations for carotid thrombectomies might not have been taken into account when the subjects were included, thus, the differences between the results of EVT in ACS and PCS might change when new guidelines are applied.

Our study aimed to compare early hospital outcomes of BAO and MCAO treated with EVT or bridging therapy (BT), and establish the factors associated with worse outcomes. To make the results of our study more applicable to modern clinical practice recommendations, we included only those MCAO subjects who were suitable for thrombectomy according to ESO-ESMINT guidelines. Thus, BAO outcomes were compared to the best outcomes possible using the modern gold standard for carotid thrombectomies. We have chosen to assess the early outcomes, as there are quite a lot of confounders not associated with the intervention itself that might have influenced the outcomes in 3 months, such as basic stroke care and nursing, and COVID-19 pandemics [25].

## 2. Materials and Methods

This retrospective single-centre study was carried out in the Neurology department of Vilnius University Hospital Santaros Klinikos (VUHSK). The research population included 279 subjects admitted to the comprehensive stroke centre from January 2015 to December 2021 due to stroke caused by large-vessel occlusion confirmed by computed tomography angiography (CTA) and treated by EVT or bridging therapy (BT) intravenous thrombolysis (IVT) + EVT). Only subjects with the large-vessel occlusion undoubtedly suitable for the mechanical thrombectomy were included, which comprised: (1) BAO group and (2) MCAO group presenting with the occlusion of the M1 segment or both M1 and M2 segments of the middle cerebral artery (MCA) that could be isolated or tandem occlusion (TO) defined as internal carotid artery (ICA) occlusion or high-grade stenosis combined with MCAO. The subgroup with TO was referred to as the MCAO-T subgroup, whereas the isolated MCAO subgroup was referred to as the MCAO-I subgroup. Subjects presenting with the occlusions of the M2 and M3 segments of the MCA, isolated vertebral artery or posterior cerebral artery (PCA) occlusion were excluded, as mechanical thrombectomy from the aforementioned vessels remains a matter of debate in some cases.

Subjects with MCAO were included according to the newest ESO-ESMINT guidelines for mechanical thrombectomy [13]. 2 subjects were excluded because of having both MCAO and BAO. 1 subject was excluded because of concomitant COVID-19 pneumonia, as COVID-19-associated stroke tends to be more severe and has worse prognosis [25].

Baseline characteristics included demographical data (sex, age), baseline NIHSS score, door-to-puncture time (DPT) and onset-to-puncture time (OPT), additionally door-to-needle time (DNT) and onset-to-needle time (ONT) in the BT subgroup, arterial recanalization according to thrombolysis in cerebral infarction (TICI) grading system (2b, 2c and 3 scores were defined as successful recanalization, whereas 0, 1 and 2a scores were defined as unsuccessful recanalization) [26], concomitant conditions (arterial hypertension (AH), heart failure classified as NYHA B or worse (HF), coronary heart disease (CHD), atrial fibrillation (AF), peripheral artery disease (PAD), which presented as lower limb PAD in all subjects, diabetes mellitus (DM), renal failure (RF), history of stroke or transient ischemic attack (TIA).

Early in-hospital mortality was selected as the primary outcome. NIHSS score after 24 h, early neurological deterioration defined as delta NIHSS > 30%, futile recanalization (FR), and ambulatory functional outcome at discharge, defined as mRS 0–3, were selected as the secondary outcomes. The frequency of complications was also assessed, including symptomatic intracranial haemorrhage (sICH), infectious complications, and recurrent ischemic stroke. sICH was defined as parenchymal haemorrhage 2 (PH-2) according to ECASS-II criteria [27] or massive subarachnoid haemorrhage associated with neurological decline.

Data were extracted from electronic patient records. If any uncertainty regarding radiological findings occurred, the CT scans were analysed by a neuroradiologist with a special interest in stroke (G.L.). Statistical analysis was performed with IBM SPSS Statistics for Windows, Version 25.0. (Armonk, NY: IBM Corp.) and R-Commander (R version 4.2.1) [28]. The Shapiro–Wilk test was used to check for normal distribution. All quantitative variables except ONT were non-normally distributed. Quantitative variables were reported as mean ± SD or median (interquartile range), as appropriate. Numeric ordinal variables were reported as median (interquartile range). To compare means between two groups, independent samples t-test was applied, and to compare medians between two groups Wilcoxon test was applied. To compare between three groups ANOVA and Kruskal–Wallis tests were applied, respectively. Nominal variables were analysed by the Chi-square test or Fisher exact test, as appropriate. *p* = 0.05 was considered the level of statistical significance. The study power for the Goodness-of-Fit test that was used to determine the primary outcome was 0.9, calculated with G-Power 3.1.

## 3. Results

Of 279 subjects, 162 (58.1%) were female. The median age was 70 (14.5) years. 224 (80.3%) presented with MCAO, of which 188 (67.4%) were isolated and 36 (12.9%) were TO, and 55 (19.7%) presented with BAO. The median NIHSS on admission was 14 (9). In 208 (74.5%) EVT only was performed, and in 71 (25.5%), BT was applied. The median DPT and OPT were 83 (37.25) min and 210 (195) min, respectively. In the BT subgroup, median DNT and mean ONT were 38.0 (23.0) min and 125.5 ± 38.7 min, respectively. 

Baseline characteristics in BAO and MCAO groups are presented in Table 1. BAO group was characterized by higher baseline NIHSS compared to the MCAO group (19 vs. 14, *p* < 0.001), and longer DPT (93 vs. 82 min, *p* = 0.034). Other baseline parameters did not differ between groups. A similar comparison was performed for three groups, distinguishing MCAO-I and MCAO-T subgroups, as the stroke mechanisms could differ in the subgroups. The analysis did not reveal any substantial differences except baseline NIHSS, which was different mainly between BAO and both MCAO subgroups (Table 2). 

Baseline characteristics did not differ significantly depending on the presence of tandem occlusion in the MCAO group. Further comparison of the outcomes was made between the BAO and MCAO groups without distinguishing MCAO-I and MCAO-T subgroups. Hospital mortality in the BAO group was almost twice as high as in the MCAO group (20.0% vs. 10.3%, *p* = 0.048), as was the frequency of FR. Other outcomes reflecting neurological and functional changes during hospitalization did not differ significantly. The frequency of recurrent stroke, sICH and infectious complications did not differ significantly between the groups, as well (Table 3).

The relationships between baseline characteristics, complications, and early in-hospital mortality are presented in Table 4. Unsuccessful recanalization, sICH, PAD and RF were significantly associated with higher early hospital mortality risk. Recurrent stroke was not included, as no recurrent strokes were observed in the BAO group.

In the BAO group, unsuccessful recanalization appeared to be the only significant predictor of hospital mortality (OR = 24.6; 95% CI = [3.10–520.75]; *p* = 0.007). Non-significant trends were observed for RF and PAD (OR 9.6; 95% CI = [0.83–219.27]; *p* = 0.078, for both risk factors). A trend for higher risk of sICH was observed in the BAO group when BT was applied (OR—5.72, 95% CI = [0.64–52.21]; *p* = 0.066), whereas in the MCAO group no significant risk was observed. Nevertheless, neither sICH nor BT was associated with increased early mortality in the BAO group.

## 4. Discussion

Our findings reconfirm the fact that the mortality in BAO is higher than in MCAO, and unsuccessful recanalization is the strongest predictor of poor outcomes [4,17,18]. Nevertheless, some new insights can be obtained. First of all, this study estimates the early hospital mortality that indicates the frequency of undoubtedly futile recanalization in most cases, while outcomes within 3 months, especially in 2019–2022 (during the COVID-19 pandemic), might have been influenced not only by EVT, but by basic stroke care, rehabilitation and nursing accessibility, and COVID-19 infection after stroke [25]. 

There was no difference in recanalization rates, early neurological dynamics, median NIHSS after 24 h, complication rates and early ambulatory outcomes between the groups. The rates of reperfusion in our study were comparable to the previously reported, and the rates of the ICH were slightly higher, which can be attributed to different ICH definitions used by different authors [29,30,31]. Those two factors were the strongest risk factors for early mortality, and they occurred at the same frequency in both groups. Nevertheless, early mortality was still two times higher in the BAO group. 

Concerning that the MCAO group was scrutinized according to the latest guidelines, these findings might be interpreted as twice as high amount of futile recanalization in BAO compared to MCAO “gold standard”. That means that some prognostic factors remain under-recognized or neglected when the decision for EVT in BAO is made. There is a strong need to identify specific prognostic factors to formulate recommendations for endovascular treatment in BAO and implement them into clinical practice.

Although posterior circulation ischemic changes and posterior collateral status are proven risk factors for END and worse prognosis [32,33], those criteria are not included in major guidelines. That may contribute to uncertainty while making clinical decisions and result in a higher number of heroic interventions.

We found a trend for higher risk of ICH when BT was applied in BAO, though the significance of this finding is limited by the small sample size. When EVT only is applied in the posterior circulation, the risk of sICH is comparable to the anterior circulation, whereas IVT only poses two times higher risk of sICH in the anterior circulation [1,34]. One study that included not only BAO but also PCA occlusion found, that BT was associated with an 8.7 higher risk of the sICH, compared to IVT only [35].

BT has been remaining a hot topic for many years. It is still unclear which subgroup would benefit from it and how it should be distinguished. The majority of the trials assessing BT investigate ACS subjects only or do not analyse the PCS subgroup separately [36,37]. A meta-analysis comparing BT to standard medical treatment specifically in PCS did reveal a significant increase in sICH, but not in 90-day mortality [38]. Unfortunately, standard medical treatment was defined as any antithrombotic treatment including antiplatelets, IVT or both, making results not exactly applicable to this discussion. It would be useful to investigate the aspect of BT in PCS further.

The limitations of our study are the retrospective design of the study and the relatively small number of BAO cases that resulted in the insufficient significance of some findings. The strengths are rigid and modern inclusion criteria based on the newest ESO-ESMINT guidelines, careful additional scrutiny of the CT findings by the radiologist, and high study power. The results of BT in BAO are also worth further investigation.

Other findings associated with the risk of poor outcomes were RF and PAD. RF is a well-known risk factor for a worse stroke prognosis after EVT and for any ICH in PCS [37,38,39]. As for PAD, we suggest that this factor was linked to an increased risk of poor outcomes because it is a marker of an unfavourable cardiovascular profile and is linked to increased mortality rates in general [40].

## 5. Conclusions

The outcomes of the EVT in BAO remain poor, compared to the modern gold standard in the ACS, resulting in twice as high hospital mortality and FR. Predictors of poor outcome are unsuccessful recanalization, sICH, PAD and RF. Indications for BT in PCS might differ from the ACS, as it could pose a higher risk for sICH. These findings highlight the necessity of clear recommendations for RT in PCS. Data available from observational studies need to be confirmed by meta-analyses and randomized controlled trials. This would decrease the number of futile interventions and help clinicians to make the informed and evidence-based decisions for the reperfusion.

## Figures and Tables

**Table 1 medicina-59-00096-t001:** Baseline characteristics of the study population in BAO and MCAO groups.

	BAO (*n* = 55)	MCAO (*n* = 224)	*p*-Value
Age (years)	71 (13)	70 (15.3)	0.480
Sex (female)	29 (52.7)	133 (59.4)	0.371
AH	48 (87.3)	193 (87.2)	0.829
AF	24 (43.6)	122 (54.5)	0.150
DM	14 (25.5)	44 (19.6)	0.341
History of stroke/TIA	13 (23.6)	45 (20.1)	0.561
CHD	13 (23.6)	72 (32.1)	0.219
PAD	3 (5.5)	14 (6.0)	0.881
HF	18 (32.7)	72 (32.3)	0.950
RF	3 (5.5)	14 (6.2)	0.825
Baseline NIHSS	21 (20)	14 (8)	<0.001 *
DPT (min)	93 (39)	82 (37)	0.034 *
OPT (min)	252.5 (267.5)	205 (190)	0.259
DNT (min)	41 (12)	37 (24)	0.291
ONT (min)	125.4 ± 31.1	125.5 ± 40.0	0.997
Successful recanalization	50 (90.9)	202 (90.2)	0.870
IVT applied	10 (18.2)	61 (27.2)	0.167

Quantitative and numeric ordinal data are presented as mean ± standard deviation or median (IQR). Qualitative data are presented as an absolute number (percentage). * Significant differences are denoted with an asterisk. BAO—basilar artery occlusion, MCAO—middle cerebral artery occlusion, DM—diabetes mellitus, AH—arterial hypertension, AF—atrial fibrillation, TIA—transient ischemic attack, MI—myocardial infarction, HF—heart failure, RF—renal failure, CHD—coronary heart disease, PAD—peripheral artery disease, NIHSS—National Institutes of Health Stroke Scale, DPT—door-to-puncture time, OPT—onset-to-puncture time, DNT—door-to-needle time, ONT—onset-to-needle time, IVT—intravenous thrombolysis.

**Table 2 medicina-59-00096-t002:** Baseline characteristics of the study population in BAO, MCAO-I, and MCAO-T subgroups.

	BAO (*n* = 55)	MCAO-I (*n* = 195)	MCAO-T (*n* = 39)	*p*-Value
Age (years)	71 (13)	70 (16)	68.5 (12.8)	0.576
Sex (female)	29 (52.7)	115 (61.2)	18 (50)	0.309
AH	48 (87.3)	163 (86.7)	30 (83.3)	0.845
AF	24 (43.6)	105 (55.9)	17 (47.2)	0.226
DM	14 (25.5)	33 (17.6)	11 (30.6)	0.135
History of stroke/TIA	13 (23.6)	38 (20.2)	7 (19.4)	0.840
CHD	13 (23.6)	60 (30.9)	12 (33.3)	0.464
PAD	3 (5.5)	10 (5.3)	2 (5.6)	0.998
HF	18 (32.7)	57 (30.5)	15 (41.7)	0.421
RF	3 (5.5)	11 (5.9)	3 (8.3)	0.829
Baseline NIHSS	21 (20)	14 (8)	14.5 (10.5)	0.003 *
DPT (min)	93.0 (39)	82 (36)	80.5 (38.7)	0.092
OPT (min)	252.5 (267.5)	205 (195)	205 (145)	0.501
DNT (min)	41 (12)	37.5 (25)	35 (19.0)	0.561
ONT (min)	125.4 ± 31.1	125.5 ± 40.3	125.5 ± 40.9	1
Successful recanalization	50 (90.9)	169 (89.9)	33 (91.7)	0.934
IVT applied	10 (18.2)	50 (26.6)	11 (30.6)	0.340

Quantitative and numeric ordinal data are presented as mean ± standard deviation or median (IQR). Categorical data are presented as an absolute number (percentage). * Significant differences are denoted by an asterisk. BAO—basilar artery occlusion, MCAO-I—isolated middle cerebral artery occlusion, MCAO-T—tandem occlusion of the internal carotid artery and middle cerebral artery, NIHSS—National Institutes of Health Stroke Scale, DM—diabetes mellitus, AH—arterial hypertension, AF—atrial fibrillation, TIA—transient ischemic attack, MI—myocardial infarction, HF—heart failure, RF—renal failure, CHD—coronary heart disease, PAD—peripheral artery disease, DPT—door-to-puncture time, OPT—onset-to-puncture time, DNT—door-to-needle time, ONT—onset-to-needle time, IVT—intravenous thrombolysis.

**Table 3 medicina-59-00096-t003:** Outcomes and complications in BAO and MCAO groups.

	BAO (*n* = 55)	MCAO (*n* = 224)	*p*-Value
Mortality	11 (20)	23 (10.3)	0.048 *
NIHSS after 24 h	7 (16.5)	6 (11)	0.243
Delta NIHSS	5 (14)	5 (10)	0.064
END	7 (12.7)	24 (10.7)	0.670
FR	11 (20)	23 (10.3)	0.048 *
Ambulatory at discharge (mRS 0–3)	32 (58.2)	123 (54.0)	0.578
Recurrent ischemic stroke	0 (0)	3 (1.3)	0.388
sICH	6 (10.9)	26 (11.6)	0.884
Infection	21 (38.3)	89 (39.7)	0.833

Quantitative and numeric ordinal data are presented as mean ± standard deviation or median (IQR). Categorical data are presented as an absolute number (percentage). * Significant differences are denoted by an asterisk. BAO—basilar artery occlusion, MCAO—middle cerebral artery occlusion, NIHSS—National Institutes of Health Stroke Scale, END—early neurological deterioration, FR—futile recanalization, mRS—modified Rankin Scale, sICH—symptomatic intracranial haemorrhage.

**Table 4 medicina-59-00096-t004:** Baseline factors, complications, and their association with the risk of early hospital mortality for the whole study population.

Factor	OR	95% CI
Age > 80	0.56	0.16–1.51
BAO	2.18	0.96–4.72
AH	2.76	0.79–17.47
AF	0.90	0.44–1.85
DM	2.01	0.89–4.34
History of stroke/TIA	1.71	0.74–3.73
CHD	1.97	0.94–4.09
PAD	4.05	1.19–12.27 *
HF	2.04	0.98–4.23
RF	3.35	1.01–9.75 *
IVT applied (BT)	0.73	0.28–1.68
Unsuccessful recanalization	8.36	3.45–20.19 *
sICH	5.10	2.14–11.80 *
Infection	0.94	0.44–1.95

* Significant factors are denoted by an asterisk. OR—odds ratio; 95% CI—95% confidence interval; BAO—basilar artery occlusion; AH—arterial hypertension; AF—atrial fibrillation; DM—diabetes mellitus; TIA—transient ischemic attack; CHD—coronary heart disease; PAD—peripheral artery disease; HF—heart failure; RF—renal failure; IVT—intravenous thrombolysis; BT—bridging therapy; TICI—thrombolysis in cerebral infarction grading system; sICH—symptomatic intracranial haemorrhage.

## Data Availability

The data presented in this study are available on request from the corresponding author. The data are not publicly available due to due to privacy restrictions.

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
