# Peer review of "Endovascular Treatment of Basilar Artery Occlusion: What Can We Learn from the Results?"

_medicina, 2022, doi:10.3390/medicina59010096_

Round 1
Reviewer 1 Report
Overall I think this paper is important and it should be included in this journal because we need as much data as possible on BAO outcomes after IV thrombolysis and thrombectomy.
I am interested in all the data presented and they are good / interesting / worthwhile.
The main criticism I have of this study is that the authors did not present any imaging criteria in the BAO patients that could be used to stratify which pt were at risk of poor outcome. Eg: length of clot on CTA, or collaterals, or size / any early infarct changes in the posterior fossa on plan CT, or posterior fossa ASPECTS score. If the authors have these to hand then it would be very good to include. If not - OK. But this is a weakness of this study.
A stylistic point here - I do not think the data necessarily support the title "do we perform too many heroic interventions?". I would humbly suggest considering a more neutral title here. BAO is a bad disease, and definitely the authors' data sugges that some patients benefitted from BAO intervention. The NIHSS after intervention on average improved, and it seems, tended to improve more than in MCAO (although this trend was not significant). If some patients had a futile intervention, then so be it, but in this bad disease a futile intervention is not that bad if some patients benefit.
I would also change the lines 57-58 "allowed to claim that 57 EVT can be effective in BAO within 12 hours at least in Asian population" to be something a little more neutral / positive, in keeping with the actual message of the study quoted.
I would also consider changing lines 50-51 "The latest European Stroke Organisation (ESO) - European Society for Minimally Invasive Neurological Therapy (ESMINT) Guidelines on Mechanical Throm- 49 bectomy in Acute Ischemic Stroke (further referred to as “ESO-ESMINT guidelines”) provide no specific EVT recommendations in PCS [13]," because the ESMINT guidelines do actually strongly recommend considering intervention in BAO, based on anterior circulation data.
Author Response
We would like to thank you for the revisions.
* I agree that the scarcity of radiological findings is a serious drawback. As in this study some data were quite old, we have only the information about native ischemia and core on CT perfusion, both of them were not associated with the poor outcome.
* As for the title (though I really liked the old one) would you find the correction to "Endovascular treatment of basilar artery occlusion: what can we learn from the results?" appropriate?
* 57-58 lines corrected --> recent results from Endovascular Treatment for Acute Basilar Artery Occlusion (ATTENTION) trial proved that EVT can be effective in BAO within 12 hours
* 50-51 lines corrected --> The latest EVT recommendations for PCS provided by European Stroke Organisation (ESO) - European Society for Minimally Invasive Neurological Therapy (ESMINT) Guidelines on Mechanical Thrombectomy in Acute Ischemic Stroke (further referred to as “ESO-ESMINT guidelines”) are based on ACS treatment results.
Reviewer 2 Report
I would like to congratulate the authors on their work! This is potentially significant research regarding the early hospital outcome of BAO and MCAO treated with EVT or bridging therapy. This is a well written article, with very powerful statistic levels. I have only minor recommendation for the authors, please explain all the abbreviation in the abstract.
Well done.
Author Response
Thank you so much!
All the abbreviations ar corrected.